# Prognostic Role of CSF β-amyloid 1–42/1–40 Ratio in Patients Affected by Amyotrophic Lateral Sclerosis

**DOI:** 10.3390/brainsci11030302

**Published:** 2021-02-27

**Authors:** Tiziana Colletti, Luisa Agnello, Rossella Spataro, Lavinia Guccione, Antonietta Notaro, Bruna Lo Sasso, Valeria Blandino, Fabiola Graziano, Caterina Maria Gambino, Rosaria Vincenza Giglio, Giulia Bivona, Vincenzo La Bella, Marcello Ciaccio, Tommaso Piccoli

**Affiliations:** 1ALS Clinical Research Center and Laboratory of Neurochemistry, Department of Biomedicine, Neuroscience and Advanced Diagnostics, University of Palermo, 90129 Palermo, Italy; tizianacolletti@gmail.com (T.C.); lavyguccione@hotmail.it (L.G.); antonietta.notaro@libero.it (A.N.); vincenzo.labella@unipa.it (V.L.B.); 2Institute of Clinical Biochemistry, Clinical Molecular Medicine and Laboratory Medicine, Department of Biomedicine, Neurosciences and Advanced Diagnostics, University of Palermo, 90127 Palermo, Italy; luisa.agnello@unipa.it (L.A.); bruna.losasso@unipa.it (B.L.S.); cmgambino@libero.it (C.M.G.); rosaria.vincenza.giglio@alice.it (R.V.G.); giulia.bivona@unipa.it (G.B.); marcello.ciaccio@unipa.it (M.C.); 3IRCSS Bonino Pulejo, Presidio Pisani, 90129 Palermo, Italy; rossellaspataro@libero.it; 4Unit of Neurology, Department of Biomedicine, Neurosciences and Advanced Diagnostics, University of Palermo, 90129 Palermo, Italy; valeriabl@libero.it (V.B.); fabiolagr93@gmail.com (F.G.)

**Keywords:** ALS, biomarker, beta amyloid

## Abstract

The involvement of β-amyloid (Aβ) in the pathogenesis of amyotrophic lateral sclerosis (ALS) has been widely discussed and its role in the disease is still a matter of debate. Aβ accumulates in the cortex and the anterior horn neurons of ALS patients and seems to affect their survival. To clarify the role of cerebrospinal fluid (CSF) Aβ 1–42 and Aβ 42/40 ratios as a potential prognostic biomarker for ALS, we performed a retrospective observational study on a cohort of ALS patients who underwent a lumbar puncture at the time of the diagnosis. CSF Aβ 1–40 and Aβ 1–42 ratios were detected by chemiluminescence immunoassay and their values were correlated with clinical features. We found a significant correlation of the Aβ 42/40 ratio with age at onset and Mini Mental State Examination (MMSE) scores. No significant correlation of Aβ 1–42 or Aβ 42/40 ratios to the rate of progression of the disease were found. Furthermore, when we stratified patients according to Aβ 1–42 concentration and the Aβ 42/40 ratio, we found that patients with a lower Aβ 42/40 ratio showed a shorter survival. Our results support the hypothesis that Aβ 1–42 could be involved in some pathogenic mechanism of ALS and we suggest the Aβ 42/40 ratio as a potential prognostic biomarker.

## 1. Introduction

Amyotrophic lateral sclerosis (ALS) is the most common degenerative motor neuron disease, which results in progressive muscle weakness and causes death in a few years. The pathogenesis of ALS is not fully understood and several pathological processes have been proposed such as abnormal protein aggregation, mitochondrial dysfunction and oxidative stress [1]. To date, the diagnosis of ALS is on clinical features and electrophysiological parameters, indicating the degeneration of both upper and lower motor neurons [2]. Heterogeneity in terms of clinical presentation often makes an early and accurate diagnosis a real challenge for clinicians. For this reason, there has been a growing interest in identifying candidate biomarkers for ALS, which can help make an early diagnosis and predict disease progression. Among these, the role of a neurofilament (NF) phosphorylated heavy chain (pNF-H) and light chain (NF-L) as potential biomarkers for ALS is defining. NFs have a non-specific and not fully clarified role in the pathogenesis of ALS. The abnormal accumulation of NF aggregates was observed in perycaria and proximal axons of motoneurons both in ALS murine models and patients that seemed to be related to an impairment of intracellular transport [3]. Recently, a few authors have shown that the aggregation of NFs is related to their hyperphosphorylation state [4]. pNF-H and NF-L are increased in the cerebrospinal fluid (CSF) of ALS patients in comparison with control groups [5,6] and the higher levels are associated with a more rapidly evolving disease and shorter survival [7]. The role of other candidate biomarkers (such as Tau proteins) is still under investigation [8,9,10].

ALS shares common pathways with other neurodegenerative disorders. For example, C9 or f72 repeat expansions and TAR DNA-binding protein (TARDBP) mutations have been described in ALS and frontotemporal lobar degeneration (FTLD), modifying the idea of ALS as a disease confined to the motor system to the extreme phenotypic expression of a clinical/pathological continuum with FTLD [11,12,13]. Furthermore, the presence of β-amyloid (Aβ) deposits at the cortical level, hippocampus and spinal cord motor neurons have been described in ALS patients [14,15,16], suggesting the possibility of some overlapping features between ALS and Alzheimer’s disease (AD).

AD is the most common cause of dementia, characterized by Aβ and Tau deposition, respectively, in senile plaques and neurofibrillary tangles as a result of a complex mechanism known as the amyloid cascade [17]. The amyloid precursor protein (APP) is processed by α-secretase into a soluble form α of the APP (sAPPα) and carbossi-terminal fragment α (CTFα) and by β-secretase sAPPβ and CTFβ. Subsequently, CTFβ is cleaved into Aβ 1–40 or Aβ 1–42 by γ-secretase and the imbalance of this process leads to an overexpression of Aβ 1–42 that precipitates, forming the senile plaques. The consequence is the hyperphosphorylation of the Tau protein and the formation of neurofibrillary tangles [18]. CSF Aβ 1–42 levels combined with total Tau (tTau) and phosphorylated Tau (pTau) are currently used as diagnostic biomarkers for AD with a high sensitivity and specificity [19,20,21,22], ameliorating the diagnostic accuracy in the very early stages of the disease.

Due to the pathogenetic similarities among neurodegenerative diseases, possible common pathways between AD and ALS have been investigated. Preclinical studies demonstrated the interaction between superoxide dismutase (SOD) and Aβ and evidence of the amyloid cascade has been reported [23] with an increase of sAPP in the CSF from ALS patients [24] and the post-mortem evidence of the over-expression of APP and Aβ in the hippocampi of ALS patients [17]. On the other hand, it is known that APP regulates glial cell-derived neurotrophic factor (GDNF) expression, having a role on neuromuscular junction formation and probably also in neuromuscular degenerative diseases [23]. 

Whether or not Aβ has a role in the pathogenesis of ALS is far from being clear but it has been recently proposed that the CSF Aβ 1–42 protein concentration is higher in ALS patients and that it is related to disease severity at the time of diagnosis [25]. 

Our aim is to evaluate the role of the CSF Aβ 1–42 and Aβ 1–40 concentration and the Aβ 42/40 ratio as a potential predictor factor for progression and overall survival in ALS.

## 2. Patients and Methods

### 2.1. Patients 

Ninety-three (93) ALS patients (M/F: 1.11) were enrolled from the ALS Clinical and Research Center, Department of Biomedicine, Neuroscience and advanced Diagnostics (Bi.N.D.), University of Palermo, Italy, from January 2001 to October 2020. All ALS patients were diagnosed according to El-Escorial revised criteria [2] combined with the neurophysiological ones [26]. The revised ALS Functional Rating Scale (ALSFRS-R) [27] was used to score the severity of the symptoms of ALS patients; a higher score indicated normality and a lower score defined a locked-in condition. ΔFS ((ALSFRS-R at onset–ALSFRS-R at time of diagnosis)/diagnostic delay) was used to define the disease progression [28]. According to the ΔFS, patients could be classified in three groups: slow progression (ΔFS < 0.5), intermediate progression (ΔFS ≥ 0.5 < 1) and rapid progression (ΔFS ≥ 1) [28]. We considered co-morbidities for each patient.

All patients underwent a cognitive/behavioral assessment and the administration of neuropsychological tests such as the Frontal Systems Behavioral Scale (FrSBe), Mini Mental State Examination (MMSE) and Edinburgh Cognitive and Behavioral ALS Screen (ECAS) (S-TAB.1). Fewer than 30% showed some degree of behavioral/cognitive impairment according to the Italian Validation of ECAS but none of them were demented. All patients were tested for the most common ALS-related genes and no known mutations associated with ALS were detected.

ALS patients underwent a lumbar puncture (LP) and a CSF analysis as routine procedures of the diagnostic work-up. For the biomarker analysis, ALS patients were subdivided into three subgroups according to the rate of progression based on ΔFS (i.e., slow: ALS-s; intermediate: ALS-i; rapid: ALS-r). All demographic and clinical features of the selected ALS patients are shown in Table 1. None of the patients enrolled assumed any specific drug for ALS treatment at the time of the LP and all of them started riluzole immediately after the diagnosis was made. None of them participated in clinical trials.

All patients gave informed written consent. The study was approved by the local Ethics Committee. All of the clinical and biological assessments were carried out in accordance with the World Medical Association Declaration of Helsinki.

### 2.2. CSF Collection and Analytical Techniques 

All CSF samples were collected in the morning hours and then sent to the Central Hospital Laboratory for a routine analysis. For biomarker detection, the CSF samples were centrifuged in case of blood contamination, aliquoted in polypropylene tubes and stored at –80 °C within one hour until further analysis according to international guidelines [29]. The CSF routine chemical parameters are shown in Table 2.

The CSF Aβ 1–42 and Aβ 1–40 were measured by a chemiluminescent immunoassay CLEIA (Lumipulse G b-amyloid 1–40, Lumipulse G b-amyloid 1–42, Fujirebio Inc. Europe, Gent, Belgium) on a fully automatic platform (Lumipulse G1200 analyzer, Fujirebio Inc. Europe, Gent, Belgium). We used as reference cut-off for the Aβ 1–42 value and the Aβ 42/40 ratio < 650 pg/ML and < 0.055, respectively, as suggested by the manufacturer.

### 2.3. Statistical Analyses 

All statistical analyses were performed using SIGMAPLOT 12.0 software package (Systat Software Inc., San Jose, CA, USA).

A Shapiro–Wilk test was performed to test the normality of the data. We expressed demographic, clinical and biochemical variables as a median with interquartile ranges (IQR). We performed Kruskal–Wallis one way analysis of variance on ranks to compare non-parametric data, a one way ANOVA to compare parametric data and a chi-squared test to assess differences between the groups. We analyzed non-parametric data with Spearman’s rank correlation coefficient and parametric data with Pearson’s correlation coefficient, considering *p* values < 0.05 as significant. 

A survival analysis was performed with the Kaplan–Meier method and survival curves were compared with the log-rank test. Univariate and multivariate Cox regression analyses were performed to predict risk factors for overall survival.

## 3. Results

A retrospective observational study was performed on 93 ALS patients to analyze the role of Aβ 1–42, Aβ 1–40 and the Aβ 42/40 ratio as candidate biomarkers for ALS. As a few studies have shown that the CSF Aβ levels are correlated with the rate of progression [21], we stratified ALS patients into three subgroups: ALS-s (*n* = 19; M/F: 2.16), ALS-i (*n* = 31; M/F: 1) and ALS-r (*n* = 35; M/F: 1.18).

In our study, the total cohort of ALS patients had a median age at onset of 67 years. 96.6% of ALS patients were sporadic with a spinal onset in 70.3% of the whole cohort. At the time of diagnosis, ALS patients showed median values of a forced vital capacity (FVC)% of 80.5 (IQR = 54.75–93.25), of a body mass index (BMI) of 24.8 kg/m2 (IQR = 21.5–27.12) and of a ΔFS of 0.81 (IQR = 0.5–1.33). The Kruskal–Wallis one way ANOVA with the rate of progression (ΔFS) as a factor showed statistically significant differences in the age of onset (lower in the ALS-s group, *p* < 0.001), diagnostic delay (longer in the ALS-s group, *p* < 0.001) and survival (longer in the ALS-s group, *p* < 0.001); no statistically significant differences were found for the M/F ratio, education, FVC% and BMI (Table 1). The CSF biochemical profile was similar in the three subgroups (Table 2). The neuropsychological assessments with FrSBe, MMSE and ECAS showed no cognitive/behavioral impairments (Appendix A).

Analyzing data with the Shapiro–Wilk test, we found that the CSF Aβ 42/40 values were normally distributed while the Aβ 1–42 and Aβ 1–40 ones were not. As shown in Figure 1, the median values of the CSF Aβ 1–42 concentration and the mean values of the Aβ 42/40 ratio resulted above the reference cut-off (< 650 pg/mL and < 0.05, respectively). We found no significant differences among the three ALS subgroups (Aβ 1–42: *p* = 0.685; Aβ 1–40: *p* = 0.340; Aβ 42/40 ratio: *p* = 0.426).

Spearman’s correlation analyses for the CSF Aβ 1–42 and Aβ 1–40 levels showed no significant correlations (Table 3) while the Pearson’s correlation analysis showed a significant correlation of Aβ 42/40 ratio values with the age at onset (*r*^2^ = −0.274, *p* = 0.008) and MMSE scores (*r*^2^ = 0.396, *p* = 0.019) (Table 4). 

To verify if the CSF Aβ proteins could affect the survival of ALS patients, we stratified ALS patients according to the median values of Aβ 1–42, Aβ 1–40 and the Aβ 42/40 ratio, obtaining three subgroups for each analyzed protein: L-1-Q (i.e., patients with values lower than the first quartile), IQR (i.e., patients with values between the first and the third quartiles) and U-3-Q (i.e., patients with values upper than the third quartile). Only for the Aβ 42/40 ratio did the Kaplan–Meier analysis with a Holm–Sidak post-hoc test show that L-1-Q patients had a significantly shorter survival (27 (IQR: 17–41) months) in comparison with U-3-Q (39 (IQR: 26–60) months) (log-rank = 6.617; *p* = 0.037) (Figure 2). 

Interestingly, patients in the L-1-Q showed a higher median age in comparison with other subgroups (L-1-Q: 71 (66.5–75.25); IQR: 66.5 (63–71.75); U-3-Q: 65.5 (61–70.5); *p* = 0.019).

Subsequently, we performed univariate and multivariate Cox regression analyses to test the predictor role of different demographic and clinical features of ALS patients and the CSF levels of Aβ 1–42, Aβ 1–40 and the Aβ 42/40 ratio. As shown in Table 5, at the univariate regression analysis, the age at onset (*p* = 0.001), diagnostic delay (*p* = 0.001), ΔFS at diagnosis (*p* < 0.001) and Aβ 42/40 ratio (*p* = 0.026) were significantly associated with overall survival. We then considered variables that were positively related to survival at the univariate analysis for the multivariate Cox regression analysis. As shown in Table 6, the diagnostic delay (*p* =0.025), ΔFS at diagnosis (*p* = 0.032) and Aβ 42/40 ratio (*p* = 0.015) were independent predictors of overall survival. Furthermore, the multivariate Cox regression analysis was performed to investigate the role of co-morbidities in overall survival but no significant data were obtained (Appendix A).

## 4. Discussion

Our study was aimed at exploring the potential role of Aβ as a prognostic biomarker in ALS. For this purpose, we designed a retrospective observational study that included 93 patients. The CSF Aβ 1–42 and Aβ 1–40 levels and the Aβ 42/40 ratio were determined and correlated with demographic, clinical and neuropsychological features of ALS patients. 

In recent years, CSF Aβ levels have been investigated to define their role as potential diagnostic and prognostic biomarkers for ALS and many studies in this field have been reported. However, the two largest studies about this topic found contrasting results. On one hand, higher CSF Aβ 1–42 levels were associated with a poorer prognosis [25] while, on the other hand, an interesting correlation of a higher concentration in patients with better performance was found, reporting increased CSF levels compared with a control group [26].

Even though the CSF Aβ and especially the Aβ 42/40 ratio represent a specific diagnostic biomarker for AD, the idea of shared mechanisms among different neurodegenerative disorders has led many authors to investigate the role of Aβ as a potential modulator of their rate of progression and overall survival. The CSF Aβ 1–42 levels were correlated to conversion from mild cognitive impairment to dementia and the progression of cognitive deficits in AD [27] as well as with the progression of cognitive impairments in Parkinson’s disease (PD) [28]. Indeed, lower CSF Aβ 1–42 levels are related to a progressive deposition of Aβ in senile plaques at the cortical level [29]. An intracellular deposition of Aβ 1–42 was also detected in the anterior horn of motor neurons of patients affected by motor neuron disease (MND) [13] while extracellular aggregates of Aβ 1–42 were detected in the hippocampus of ALS and ALS-FTD patients [17]. Studies on murine models of ALS (i.e., SOD1 G93A mice) correlated the overexpression of Aβ with an earlier onset of motor symptoms [30]. Furthermore, few cases of co-morbidity between ALS and AD in a patient showing an overlapping clinical picture have been reported [15,25]. 

In our study, ALS patients were subdivided into three subgroups (i.e., ALS-s, ALS-i and ALS-r) to analyze the contribution of the CSF Aβ levels on the rate of progression. No statistically significant differences were detected among three subgroups for the CSF Aβ 1–42, Aβ 1–40 and the Aβ 1–42/40 ratio. Indeed, no statistically significant correlation between the Aβ and clinical features including the rate of progression was found. We found a significant correlation between the Aβ 42/40 ratio with the age at onset and MMSE scores.

When analyzing the contribution of the CSF Aβ 1–42, Aβ 1–40 and the Aβ 1–42/40 ratio on overall survival of ALS patients we found that patients with lower Aβ 42/40 ratio values showed a shorter survival in comparison with those with higher values. This finding was confirmed by univariate and multivariate Cox regression analyses, which showed that the Aβ 42/40 ratio could act as an independent predictor for overall survival for ALS patients. A decrease in the CSF Aβ 42/40 ratio values could be indicative of a decrease of the CSF Aβ 1–42 levels because it might deposit in different districts of the central nervous system (CNS) as previously described [29]. In ALS, the presence of intracellular or extracellular aggregates of Aβ 1–42 is probably related to an accumulation of APP following neuronal injury. This accumulation could be due to an impairment of axon-plasmatic transport or enhanced biosynthesis of APP, representing an early neuroprotective phase to contrast extracellular and intracellular stresses. As the neuronal injuries continue, a shift toward a neurotoxic phase can occur. APP could be subjected to cleavage in Aβ by alternative mechanisms: caspase 3 giving rise to intracellular aggregates, an accumulation of which gives an increase in oxidative stress, while β-secretase contributes to extracellular deposition [31]. All of these mechanisms may contribute to a decrease of the CSF Aβ 1–42. Studies on murine models of ALS correlated the production of Aβ by β-secretase and consequently deposition as a key event that could improve motor functions and survival [32]. For this purpose, those authors treated asymptomatic and symptomatic SOD-1 G93A mice with a monoclonal antibody able to interfere with β-secretase activity, avoiding the formation of intracellular or extracellular Aβ aggregates: treated asymptomatic ALS mice showed a delay of the onset of symptoms, motor failure and death; however, the same effects were not obtained in treated symptomatic ALS mice.

Another interesting result that we obtained was related to the evidence that ALS patients with lower Aβ 42/40 ratio values presented a higher age at onset than those with higher values. These data were enforced with the finding that there was a statistically significant correlation of Aβ 42/40 ratio values with the age at onset. Considering that the age at onset was considered a strong prognostic factor for ALS [33] and that in cognitively normal subjects the concentration of the CSF AD biomarkers, including the Aβ 42/40 ratio, is associated with age [34], the decrease of the Aβ 42/40 ratio in the CSF of ALS patients might indicate that the triggering of the Aβ cascade could represent an early event that leads to an asymptomatic form of dementia that fails to fully become symptomatic as death occurs. Thus, the coexistence of an elevated age at onset and a low CSF Aβ 42/40 could represent a more severe prognostic condition that could influence survival time. The evidence that ALS patients with a low CSF Aβ 42/40 ratio had a high median age at onset make us speculate that the intracellular or extracellular deposition of Aβ would accelerate the course of the disease, worsening their survival. Indeed, we cannot exclude that a few patients in our population could have a preclinical condition of AD or the presence of age-related amyloid deposition.

Our study had a few limitations. First, the sample size. We studied 96 patients but, for our analyses, we stratified these into three groups resulting in quite small numbers. Related to that is the difference of the age of onset among groups that could partially reduce the significance of our results. Another limitation was the lack of a follow-up for the cognitive formal evaluation. This could have been useful for a correlation with the clinical progression but we found no correlation at the baseline and none of our patients developed clinically significant dementia. Finally, the lack of a control population represented a further limitation. Our goal was to assess the role of the CSF Aβ 1–42, Aβ 1–40 and the Aβ 42/40 ratio in the clinical progression of patients affected by ALS and a comparison with a control group could have enriched our work but we did not consider this to be mandatory.

## 5. Conclusion

In our study, we aimed to evaluate the potential role of Aβ in predicting the prognosis in ALS patients. We found that Aβ 42/40 is an independent predictor for survival and could be proposed as a potential prognostic biomarker as suggested by previous reports. Further studies are needed to confirm our findings in a larger population but we consider that we have added a piece to the understanding and management of the disease.

## Figures and Tables

**Figure 1 brainsci-11-00302-f001:**
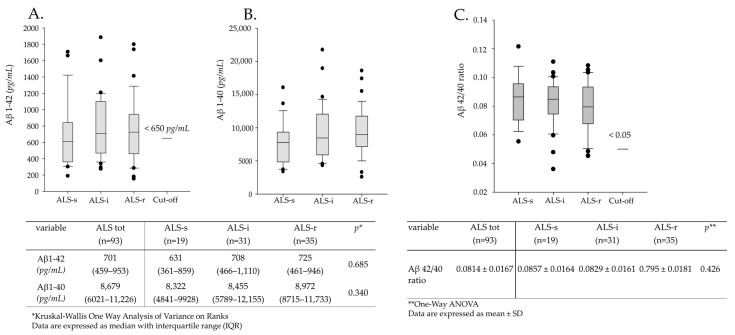
Cerebrospinal fluid (CSF) Aβ levels in ALS patients and in patients with slow (ALS-s), intermediate (ALS-i) and rapid (ALS-r) progression. (**A**) Aβ 1–42; (**B**) Aβ 1–40, (**C**) Aβ 42/40 ratio. Solid dots in A–C represent known outliers.

**Figure 2 brainsci-11-00302-f002:**
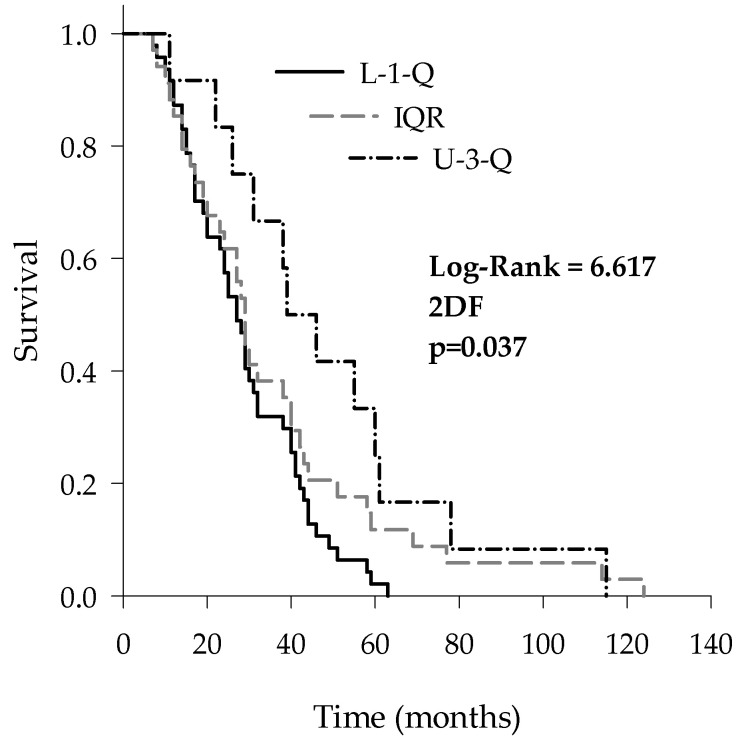
Kaplan–Meier survival curves of ALS patients stratified according to the median CSF values of the Aβ 42/40 ratio: lower than the first quartile (L-1-Q), interquartile range (IQR) and upper than the third quartile (U-3-Q).

**Table 1 brainsci-11-00302-t001:** Demographic and clinical characteristics of the total cohort and amyotrophic lateral sclerosis (ALS) patients stratified based on their rate of progression: slow (ALS-s), intermediate (ALS-i) and rapid (ALS-r). Data are expressed as a median with an interquartile range (IQR).

Variables	ALS tot (*n* = 93)	ALS-s (*n* = 19)	ALS-i (*n* = 31)	ALS-r (*n* = 35)	*p*
Age at onset (years)	67 (63–72)	63 (61–67)	67 (64–72)	70 (64–74)	< 0.001 *
M/F	1.11	2.16	1	1.18	0.43 ** χ^2^ = 1.67 with 2 DF
Education (years)	5 (5–13)	13 (5–13)	8 (5–8)	5 (5–9)	0.283 *
Type of onset familiar, % sporadic, %	3.2% 96.6%	5.2% 94.8%	3.3% 96.7%	2.8% 97.2%	0.90 ** χ^2^ = 0.22 with 2 DF
Site of onset Spinal, % Bulbar, %	70.3% 29.7%	89.5% 10.5%	63.3% 36.7%	62.3% 37.2%	0.09 ** χ^2^ = 4.80 with 2 DF
Diagnostic delay (months)	12 (9–20)	25 (18–37)	12 (10–24)	7 (4–9.5)	< 0.001 *
Rate of progression (ΔFS) ^A^	0.8 (0.5–1.3)	–	–	–	–
FVC ^a^ (%)	81 (55–93)	84 (59–98)	83 (60–92)	67 (46–93)	0.334 *
BMI ^b^ (kg/m^2^)	24.8 (21.5–27.1)	25 (21–28)	24 (22–27)	25.7 (46–83)	0.355 *
Survival (months)	30 (20–46)	57 (37–67)	35 (27–47)	17 (13–26)	< 0.001 *

^A^ ΔFS at diagnosis = (ALSFRS-R at onset–ALSFRS-R at diagnosis)/diagnostic delay (months). ^a^ Forced vital capacity. ^b^ Body mass index. * Kruskal–Wallis one way analysis of variance on ranks. ** chi-squared test. Bold font indicates a statistical significance (*p* < 0.05).

**Table 2 brainsci-11-00302-t002:** Cerebrospinal parameters in ALS patients and in patients with slow (ALS-s), intermediate (ALS-i) and rapid (ALS-r) progression. Data are expressed as a median with an interquartile range (IQR).

Parameters	ALS tot (*n* = 93)	ALS-s (*n* = 19)	ALS-i (*n* = 31)	ALS-r (*n* = 35)	*p* *
Proteins (mg/dL)	39 (28–51)	37 (19–52)	39 (32–62)	37 (26–48)	0.524
Glucose (mg/dL)	60 (55–66)	58 (55–63)	56 (51–66)	62 (57–72)	0.103
Cells (lymphocytes)	0.8 (0.6–1.8)	0.8 (0.6–2.3)	1 (0.6–2.9)	0.8 (0.4–1.6)	0.371
Oligoclonal bands (y/n)**	17/76	4/15	4/27	7/28	

* Kruskal–Wallis one way analysis of variance on ranks. ** y = yes, n = no

**Table 3 brainsci-11-00302-t003:** Spearman’s correlation of the CSF Aβ 1–42 and Aβ 1–40 with demographic, clinical and neuropsychological features of ALS patients.

	Aβ 1–42	Aβ 1–40
Age at onset (years)	*r* = −0.041, *p* = 0.695	*r* = 0.312, *p* = 0.208
Diagnostic delay (months)	*r* = −0.140, *p* = 0.189	*r* = –0.163, *p* = 0.126
Rate of progression (ΔFS)	*r* = 0.008 *p* = 0.936	*r* = 0.103, *p* = 0.347
FVC (%)	*r* = 0.141, *p* = 0.237	*r* = 0.116, *p* = 0.330
FrSBe	*r* = 0.125, *p* = 0.370	*r* = 0.185, *p* = 0.196
MMSE	*r* = −0.240, *p* = 0.146	*r* = −0.429, *p* = 0.007
ECAS	*r* = 0.005, *p* = 0.979	*r* = −0.049, *p* = 0.304
Survival (months)	*r* = 0.106, *p* = 0.933	*r* = −0.119, *p* = 0.350

Frontal Systems Behavioral Scale (FrSBe). Mini Mental State Examination (MMSE). Edinburgh Cognitive and Behavioral ALS Screen (ECAS). Bold font indicates a statistical significance (*p* < 0.05).

**Table 4 brainsci-11-00302-t004:** Pearson’s correlation of the CSF Aβ 42/40 ratio with demographic, clinical and neuropsychological features of ALS patients.

Parameters	*r* ^2^	*p*
Age at onset (years)	−0.274	0.008
Diagnostic delay (months)	0.038	0.719
ΔFS	−0.086	0.432
FVC(%)	0.198	0.095
FrSBe	−0.076	0.695
MMSE	0.396	0.019
ECAS	0.054	0.792
Survival (months)	0.164	0.196

Bold font indicates a statistical significance (*p* < 0.05).

**Table 5 brainsci-11-00302-t005:** Univariate Cox regression analysis for the overall survival for ALS patients.

Parameters	b	± SE	*p*	HR	95% CI
Gender (M vs. F)	0.073	0.251	0.772	1.075	0.658–1.757
Age at onset	0.062	0.019	0.001	1.064	1.024–1.105
Site of onset (spinal vs. bulbar)	−0.443	0.271	0.102	0.642	0.378–1.092
Diagnostic delay	−0.038	0.012	0.001	0.963	0.000–0.986
FVC%	0.006	0.006	0.308	0.994	0.982–1.006
ΔFS at diagnosis	0.443	0.105	< 0.001	1.557	1.266–1.914
Aβ 1–42	0.000	0.000	0.862	1	0.999–1.001
Aβ 1–40	0.000	0.000	0.275	1	1–1
Aβ 42/40 ratio	−18.137	8.164	0.026	1.33 × 10^–8^	0.000–0.118

b = regression coefficient; SE = standard error; HR = hazard ratio; CI = confidence interval. Bold font indicates a statistical significance (*p* < 0.05).

**Table 6 brainsci-11-00302-t006:** Multivariate Cox regression analysis for overall survival for ALS patients. Significative variables believed to be significant at the univariate analysis were considered for multivariate analysis.

Parameters	b	± SE	*p*	HR	95% CI
Age at onset	0.038	0.022	0.08	1.038	0.994–1.085
Diagnostic delay	−0.032	0.014	0.025	0.968	0.000–0.996
ΔFS at diagnosis	0.301	0.140	0.032	1.351	1.026–1.779
Aβ 42/40 ratio	−20.662	8.504	0.015	1.6 × 10^–9^	0.000–0.018

b = regression coefficient; SE = standard error; HR = hazard ratio; CI = confidence interval. Bold font indicates a statistical significance (*p* < 0.05).

## Data Availability

The data presented in this study are available on request from the corresponding author. The data are not publicly available due to current privacy laws.

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
