# Peer review of "Prognostic Role of CSF β-amyloid 1–42/1–40 Ratio in Patients Affected by Amyotrophic Lateral Sclerosis"

_brainsci, 2021, doi:10.3390/brainsci11030302_

Round 1

Reviewer 1 Report

The paper on the prognostic role of CSF β-amyloid 1-42/1-40 ratio in patients af-2 fected by Amyotrophic Lateral Sclerosis by C. Tiziana et al, is well written, scietifically sound and intersting to researchers in the field. The correlations between AD and ALS have been known for some time but there has not been any clear evidence of A-beta being a prognostic marker in ALS. That is a scientifuc path worth exploring. 
I recommend publishing the paper in a present form, with just minor typo corrections i.e line 166: significative - s/b significant; line 210: were associated to a ...s/b were associated with; line 261: might induce us..s/b might indicate that...

Author Response

  1. line 166: significative - s/b significant

We corrected the text as suggested

  1. line 210: were associated to a ...s/b were associated with;

We corrected the text as suggested

  1. line 261: might induce us..s/b might indicate that...

We corrected the text as suggested

Reviewer 2 Report

The author performed a retrospective observational study on a cohort of ALS patients to find the prognostic role of the CSF β-amyloid 1-42/1-40 ratio in patients affected by ALS. I suggest the author to perform a more statistical analysis of the data.

  • Add the role of pNF-H and NF-L in the pathophysiology of ALS.
  • Does the author check the normality of the data?
  • Give details of the FrSBe, MMSE, and ECAS at baseline and follow up.
  • Is there any association between FrSBe, MMSE, and ECAS with CSF Aβ1-40, Aβ1-42, and Aβ1-42? Give details.
  • I suggest the author to perform some regression analysis to predict the prognostic role of CSF Aβ1-40, Aβ1-42, and Aβ1-42 in the patients of ALS.

Reviewer 3 Report

Tiziana et al described CSF Aβ42 and Aβ42/40 in 93 ALS patients with various degrees of progression. While the authors didn't find correlations between Aβ42 levels or Aβ42/40 ratio and the rate of progression, they claimed that a low Aβ42/40 ratio was correlated with shorter survival and higher median age. Therefore, they concluded that Aβ42/40 could be a prognostic biomarker and that intracellular or extracellular deposition of Aβ would accelerate the course of the disease, improving their survival (?). The manuscript was clearly written. However, the conclusions were confusing and prematurely made, while the experimental design and data analysis lacked the necessary rigor to support the conclusion. The manuscript is not appropriate for publication in its current state. Here are a few major concerns:

1. There are multiple problems with subject recruitment and analysis. The sample size was too small for a study like this, the paper included only 19 slow-progressing patients with a younger age of onset (P<0.001). While the authors did include both male and female, the gender difference was not noted. Interestingly, the site of onset was also different among the 3 groups of interest, which would also predict survival, as bulbar onset is known to be more aggressive. Finally, the other comorbidities and treatment of the patients were not described at all while they could affect both the survival rate and CSF Aβ42 and 40 levels.

2. The rationale behind Aβ42/40 as a prognosis marker is not well-argued or supported. Aβ42/40 has been studied as a biomarker for AD, while the evidence and rationale are much stronger in the AD case, the results have been controversial. Moreover, many aged patients seem to have increased Aβ plagues without showing any signs of dementia. The authors here failed to provide strong evidence that Aβ and APP are directly involved in ALS pathology, particularly in ALS patients without FTD, the target population of this study. The authors also stated that 96% of cases were sporadic (although they also stated that none of the patients had genetic ALS mutations, what happened in the other 4%?), most sporadic ALS cases are complex, therefore, it is unclear whether the Aβ levels correlate with ALS pathology per se. The authors stated that the patients were not clinically demented, however, the pathology data was largely missing. 

3. Results don't support the conclusions. The only positive correlation the author claimed was the CSF Aβ42/40 ratio and survival, there is no correlation between the Aβ42 level or Aβ42/40 ratio and the rate of progression. Furthermore, the autors stated that those patients with low Aβ42/40 ratio also had higher median age, in addition to shorter survival. As age is a better indicator of survival, it is hard to convince the reviewer that this is specific for ALS, without providing data in controls without ALS. It is premature to state that Aβ42/40 is a biomarker for ALS prognosis given its poor correlation with disease progression and its lack of a direct connection to ALS survival. The authors further concluded that intracellular or extracellular deposition of Aβ would accelerate the course of the disease, improving their survival. This statement is not only unclear as the two parts of the statement contradict each other, but also unfounded as there is no supportive data from the current study.

Round 2

Reviewer 2 Report

The manuscript has been improved in the current revision and the authors addressed my previous concerns. The author analyzed all correlations were with Spearman’s Rank Correlation.

The Pearson correlation evaluates the linear relationship between two continuous variables and the Spearman correlation evaluates the monotonic relationship between two continuous or ordinal variables. Spearman correlation is often used to evaluate relationships involving ordinal variables. Please re-analyze the data for the correlation test.

Author Response

We thank the reviewer for the suggestion. We performed further analyses, using Spearman’s Rank Correlation Coefficient for non-parametric data and Pearson’s Correlation Coefficient for parametric ones. We correlated CSF Aβ1-42 and Aβ 1-40 levels and Aβ42/40 ratio values in the whole ALS cohort to demographic, clinical, and neuropsychological features and found a significant correlation of Aβ42/40 ratio values with age at onset and MMSE scores. We added these further results in the text. We also corrected English language errors.

Reviewer 3 Report

The authors have addressed most of the concerns in the revised manuscript, though the reviewer is still not convinced that Aβ42/40 serves as a prognostic biomarker for ALS. However, the data presented here is of scientific value and does provide to a certain degree additional information for studying ALS.

Author Response

We thank the reviewer for the comment. We added some additional sentences in the manuscript which, in our opinion, has been enriched by further results.